# Robust Self-Supervised Learning with Lie Groups

## Abstract

Deep learning has led to remarkable advances in computer vision. Even so, today's best models are brittle when presented with variations that differ even slightly from those seen during training. Minor shifts in the pose, color, or illumination of an object can lead to catastrophic misclassifications. State-of-the art models struggle to understand how a set of variations can affect different objects. We propose a framework for instilling a notion of how objects vary in more realistic settings. Our approach applies the formalism of Lie groups to capture continuous transformations to improve models' robustness to distributional shifts. We apply our framework on top of state-of-the-art self-supervised learning (SSL) models, finding that explicitly modeling transformations with Lie groups leads to substantial performance gains of greater than 10% for MAE on both known instances seen in typical poses now presented in new poses, and on unknown instances in any pose. We also apply our approach to ImageNet, finding that the Lie operator improves performance by almost 4%. These results demonstrate the promise of learning transformations to improve model robustness[1].

## 1 Introduction

State-of-the-art models have proven adept at modeling a number of complex tasks, but they struggle when presented with inputs different from those seen during training. For example, while classification models are very good at recognizing buses in the upright position, they fail catastrophically when presented with an upside-down bus since such images are generally not included in standard training sets (Alcorn et al., 2019). This can be problematic for deployed systems as models are required to generalize to settings not seen during training ("out-of-distribution (OOD) generalization"). One potential explanation for this failure of OOD generalization is that models exploit any and all correlations between inputs and targets. Consequently, models rely on heuristics that while effective during training, may fail to generalize, leading to a form of "supervision collapse" (Jo & Bengio, 2017; Ilyas et al., 2019; Doersch et al., 2020; Geirhos et al., 2020a). However, a number of models trained without supervision (self-supervised) have recently been proposed, many of which exhibit improved, but still limited OOD robustness (Chen et al., 2020; Hendrycks et al., 2019; Geirhos et al., 2020b).

The most common approach to this problem is to reduce the distribution shift by augmenting training data. Augmentations are also key for a number of contrastive self-supervised approaches, such as SimCLR (Chen et al., 2020). While this approach can be effective, it has a number of disadvantages. First, for image data, augmentations are most often applied in pixel space, with exceptions e.g. Verma et al. (2019). This makes it easy to, for example, rotate the entire image, but very difficult to rotate a single object within the image. Since many of the variations seen in real data cannot be approximated by pixel-level augmentations, this can be quite limiting in practice. Second, similar to adversarial training (Madry et al., 2017; Kurakin et al., 2016), while augmentation can improve performance on known objects, it often fails to generalize to novel objects (Alcorn et al., 2019). Third, augmenting to enable generalization for one form of variation can often harm the performance on other forms of variation (Geirhos et al., 2018; Engstrom et al., 2019), and is not guaranteed to provide the expected invariance to variations (Bouchacourt et al., 2021b). Finally, enforcing invariance is not guaranteed to provide the correct robustness that generalizes to new instances (as discussed in Section 2).

---

[1]Code to reproduce all experiments will be available upon acceptance.

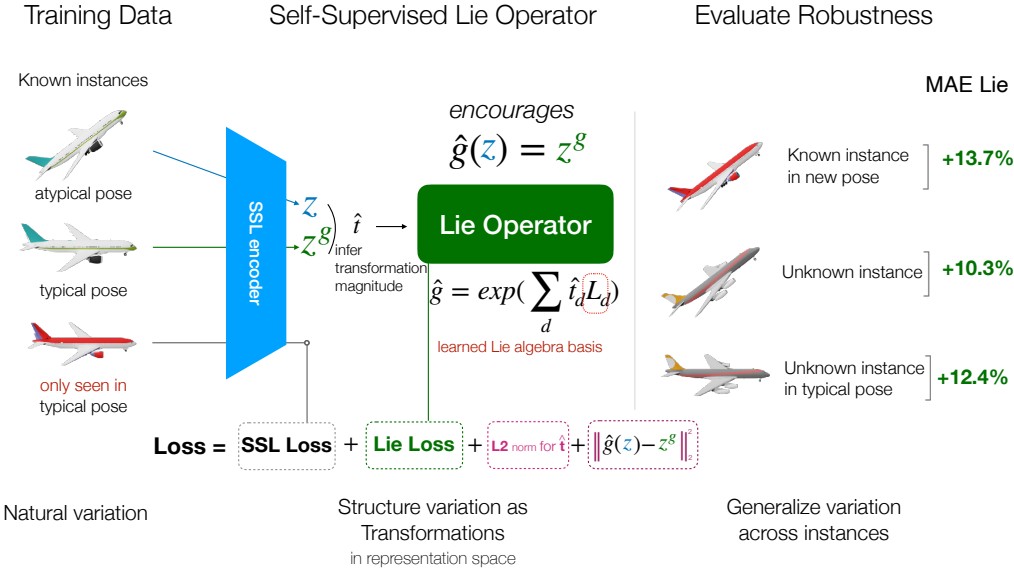

Figure 1: **Summary of approach and gains.** We generate a novel dataset containing rendered images of objects in typical and atypical poses, with some instances only seen in typical, but not atypical poses (left). Using these data, we augment SSL models such as MAE with a learned Lie operator which approximates the transformations in the latent space induced by changes in pose (middle). Using this operator, we improve performance by >10% for MAE for both known instances in new poses and unknown instances in both typical and atypical poses (right).

For these reasons, we choose to explicitly model the transformations of the data as transformations in the latent representation rather than trying to be invariant to it. To do so, we use the formalism of Lie groups. Informally, Lie groups are continuous groups described by a set of real parameters (Hall, 2003). While many continuous transformations form matrix Lie groups (e.g., rotations), they lack the typical structure of a vector space. However, Lie group have a corresponding vector space, their Lie algebra, that can be described using basis matrices, allowing to describe the infinite number elements of the group by a finite number of basis matrices. Our goal will be to learn such matrices to directly model the data variations.

To summarize, our approach structures the representation space to enable self-supervised models to generalize variation across objects. Since many naturally occurring transformations (e.g., pose, color, size, etc.) are continuous, we develop a theoretically-motivated operator, the *Lie operator*, that acts in representation space (see Fig. 1). Specifically, the Lie operator learns the continuous transformations observed in data as a vector space, using a set of basis matrices. With this approach, we make the following contributions:

1. We generate a novel dataset containing 3D objects in many different poses, allowing us to explicitly evaluate the ability of models to generalize to both known objects in unknown poses and to unknown objects in both known and unknown poses (Section 3).

2. Using this dataset, we evaluate the generalization capabilities of a number of standard models, including ResNet-50, ViT, MLP-Mixer, SimCLR, CLIP, VICReg, and MAE finding that all state-of-the-art models perform relatively poorly in this setting (Section 3.2).

3. We incorporate our proposed Lie operator in two recent SSL approaches: masked autoencoders (MAE, (He et al., 2021)), and Variance-Invariance-Covariance Regularization (VICReg (Bardes et al., 2021)) to directly model transformations in data (Section 2), resulting in *substantial OOD performance gains of greater than 10% for MAE* and of up to 8% for VICReg (Section 4.1). We also incorporate our Lie model in SimCLR (Chen et al., 2020) (Appendix E).

4. We run systemic ablations of each term of our learning objective in the MAE Lie model, showing the relevance of every component for best performance (Section 4.3).

5. We challenge our learned Lie operator by running evaluation on Imagenet as well as a rotated version of ImageNet, improving MAE performance by 3.94% and 2.62%, respectively. We also improve over standard MAE with MAE Lie when evaluated with finetuning on the iLab (Borji et al., 2016) dataset. These experiments show the applicability of our proposed Lie operator to realistic, challenging datasets (Section 5).

## 2 METHODS

Humans recognize even new objects in all sorts of poses by observing continuous changes in the world. To mimic humans' ability to generalize variation, we propose a method for learning continuous transformations from data. We assume we have access to pairs of data, e.g., images, $\mathbf{x}, \mathbf{x}'$ where $\mathbf{x}'$ is a transformed version of $\mathbf{x}$ corresponding to the same *instance*, e.g., a specific object, undergoing variations. This is a reasonable form of supervision which one can easily acquire from video data for example, where pairs of frames would correspond to pairs of transformed data, and has been employed in previous works such as Ick & Lostanlen (2020); Locatello et al. (2020); Connor et al. (2021b); Bouchacourt et al. (2021a).

Despite widespread use of such pairs in data augmentation, models still struggle to generalize across distribution shifts (Alcorn et al., 2019; Geirhos et al., 2018; Engstrom et al., 2019). To solve this brittleness, we take inspiration from previous work illustrating the possibility of representing variation via latent operators (Connor et al., 2021b; Bouchacourt et al., 2021a; Giannone et al., 2019). Acting in latent space avoids costly modifications of the full input dimensions and allows us to express complex operators using simpler latent operators. We learn to represent data variation explicitly as group actions on the model's representation space $\mathcal{Z}$. We use operators theoretically motivated by group theory, as groups enforce properties such as composition and inverses, allowing us to structure representations beyond that of an unconstrained network (such as an multi-layer perceptron, MLP) (see Appendix A). Fortunately, many common transformations can be described as groups.

We choose matrix Lie groups to model continuous transformations, as they can be represented as matrices and described by real parameters. However, they do not possess the basic properties of a vector space e.g., adding two rotation matrices does not produce a rotation matrix, without which back-propagation is not possible. For each matrix Lie group $G$ however, there is a corresponding Lie algebra $\mathfrak{g}$ which is the tangent space of $G$ at the identity. A Lie algebra forms vector space where addition is closed and where the space can be described using basis matrices. Under mild assumptions over $G$, any element of $G$ can be obtained by taking the exponential matrix map: $\exp(t\mathbf{M})$ where $\mathbf{M}$ is a matrix in $\mathfrak{g}$, and $t$ is a real number. Benefiting from the vector space structure of Lie algebras, we aim to learn a basis of the Lie algebra in order to represent the group acting on the representation space, using the $\exp$ map. We refer the reader to Appendix A and Hall (2003); Stillwell (2008) for a more detailed presentation of matrix Lie groups and their algebra.

### 2.1 LEARNING TRANSFORMATIONS FROM ONE SAMPLE TO ANOTHER VIA LIE GROUPS

We assume that there exists a ground-truth group element, $g$, which describes the transformation in latent space between the embedding of $\mathbf{x}$ and $\mathbf{x}'$. Our goal is to approximate $g$ using its Lie algebra. More formally, given a pair of samples (e.g., a pair of video frames), $\mathbf{x}$ and $\mathbf{x}'$, and an encoder, $f(\cdot)$, we compute the latent representations $\mathbf{z}$ and $\mathbf{z}^g$ as $\mathbf{z} = f(\mathbf{x})$ and $\mathbf{z}^g = f(\mathbf{x}')$. We therefore assume $\mathbf{z}^\mathbf{g} = f(\mathbf{x}') = g(f(\mathbf{x}))$. We aim to learn an operator $\hat{g}$ maximizing the similarity between $\mathbf{z}^{\hat{g}} = \hat{g}(f(\mathbf{x}))$ and $\mathbf{z}^g$.

We construct our operator by using the vector space structure of the group's Lie algebra $\mathfrak{g}$: we learn a basis of $\mathfrak{g}$ as a set of matrices $\mathbf{L_k} \in \mathbb{R}^{m \times m}, k \in \{1, \ldots, d\}$ $m$ is the dimension of the embedding space $\mathcal{Z}$ and $d$ is the dimension of the Lie algebra, i.e. the number of Lie generator matrices. Intuitively, $d$ expresses the degrees of freedom required to model the transformations. Specifically, it corresponds to the dimension of the Lie algebra as a vector space[2]. We use this basis as a way to represent the group elements, and thus to model the variations that data samples undergo. A matrix $\mathbf{M}$ in the Lie algebra $\mathfrak{g}$ thus writes $\mathbf{M} = \sum_{k=1}^{d} t_k \mathbf{L_k}$ for some coordinate vector

---

[2]In our preliminary experiments, we have cross-validated between d=1 and d=5 and found it did not seem to alter the results, hence we use d=1 in our reported results.

$\mathbf{t} = (t_1, \ldots, t_d) \in \mathbb{R}^d$. Let us consider a pair $\mathbf{z}, \mathbf{z}^g$ with $\mathbf{z}^g = g(\mathbf{z})$. We assume the exponential map is surjective[3], thus, there exists $\sum_{k=1}^d t_k \mathbf{L_k} \in \mathfrak{g}$ such that

$$g = \exp(\sum_{k=1}^d t_k \mathbf{L_k}) = \exp(\mathbf{t}^\mathsf{T} \mathbf{L}) \tag{1}$$

where $\mathbf{t} = (t_1, \ldots, t_d)$ are the coordinates in the Lie algebra of the group element $g$, and $\mathbf{L}$ is the 3D matrix in $\mathbb{R}^{d \times m \times m}$ concatenating $\{\mathbf{L_k}, k \in \{1, \ldots, d\}\}$.

Our goal is to infer $\hat{g}$ such that $\hat{g}(\mathbf{z})$ (the application of the inferred $\hat{g}$ to the representation $\mathbf{z}$) equals $\mathbf{z}^g$. We denote $\mathbf{z}^{\hat{g}} = \hat{g}(\mathbf{z})$. Specifically, our transformation inference procedure consists of Steps 1-4 in Algorithm 1. For step 1 in Algorithm 1, similar to Connor et al. (2021b); Bouchacourt et al. (2021a), we use a pair of latent codes to infer the latent transformation. Additionally, we consider that we can access a scalar value, denoted $\delta_{x,x'}$, measuring "how much" the two samples differ. For example, considering frames from a given video, this would be the number of frames that separates the two original video frames $\mathbf{x}$ and $\mathbf{x}'$, and hence comes *for free*[4].The coordinate vector corresponding to the pair $(\mathbf{z}, \mathbf{z}^g)$ is then inferred using a a multi-layer perceptron $h$[5]:

$$\hat{\mathbf{t}} = h(\mathbf{z}, \mathbf{z}^g, \delta_{\mathbf{x},\mathbf{x}'}) \tag{2}$$

When $\delta_{\mathbf{x},\mathbf{x}'}$ is small, we assume the corresponding images and embedding differ by a transformation close to the identity which corresponds to an infinitesimal $\hat{\mathbf{t}}$. To illustrate this, we use a squared $L_2$-norm constraint on $\hat{\mathbf{t}}$. For smaller $\delta_{\mathbf{x},\mathbf{x}'}$ (e.g., close-by frames in a video), larger similarity between $\mathbf{z}, \mathbf{z}^g$ is expected, in which case we want to encourage a smaller norm of $\hat{\mathbf{t}}$:

$$s(\mathbf{z}, \mathbf{z}^g, \delta_{\mathbf{x},\mathbf{x}'})||\hat{\mathbf{t}}||^2 \tag{3}$$

where $s : \mathcal{Z} \times \mathcal{Z} \times \mathbb{R} \to [0, 1]$ measures the similarity between $\mathbf{z}$ and $\mathbf{z}^g$. In our experiments, we simply use $s(\mathbf{z}, \mathbf{z}^g, \delta_{\mathbf{x},\mathbf{x}'})$ as a function of $\delta_{\mathbf{x},\mathbf{x}'}$ and compute

$$s(\delta_{\mathbf{x},\mathbf{x}'}) = \frac{1}{1 + \exp(|\delta_{\mathbf{x},\mathbf{x}'}|)}. \tag{4}$$

When $|\delta_{\mathbf{x},\mathbf{x}'}|$ decreases, $s$ increases which strengthens the constraint in Eq. 3: the inferred vector of coordinates $\hat{\mathbf{t}}$ is therefore enforced to have small norm.

## 2.2 LIE LEARNING OBJECTIVE

We encourage $\mathbf{z}^{\hat{g}}$ to be close to $\mathbf{z}^g$, while preventing a collapse mode where every transformation would be inferred as identity. In practice, recall we obtain $\mathbf{z}^g$ as the embedding $f(\mathbf{x}')$ of a corresponding frame. Specifically, we use an InfoNCE loss to match pairs[6]:

$$L_{\text{Lie}}(\mathbf{z}^{\hat{g}}, \mathbf{z}^g) = -\log \frac{\exp(sim(\mathbf{z}^{\hat{g}}, \mathbf{z}^g))}{\exp(sim(\mathbf{z}^{\hat{g}}, \mathbf{z}^g)) + \sum_{\tilde{\mathbf{z}} \in \mathcal{N}} \exp(sim(\tilde{\mathbf{z}}, \mathbf{z}^g))} \tag{5}$$

where $sim$ is a similarity function measuring how close two latent embeddings are (e.g., cosine similarity in our experiments), and $\tilde{\mathbf{z}}$ belongs to set of negatives $\mathcal{N}$ for $\mathbf{z}^g$. For a given $\mathbf{z}^g$ of a given instance, the set $\mathcal{N}$ is comprised of latent codes of other instances. We also include other instances' transformed versions and their corresponding inferred transformed embeddings as negative samples.

Crucially, we also include in the negatives set of $\mathbf{z}^g$ (denoted $\mathcal{N}$) the frame encoding $\mathbf{z}$ in order to prevent the model from collapsing to a point where all the encodings of the same instance (i.e. any transformed version) would be mapped to the same point, and the inferred $\hat{g}$ would simply learned as identity. Indeed, one can view transformed pairs simply as another form of data augmentation, such that enforcing that $\mathbf{z}^g$ matches $\mathbf{z}$ (i.e. implicitly enforcing invariance) would

---

[3]This is the case if $G$ is compact and connected (Hall, 2003, Corollary 11.10).

[4]Assuming that nearby video frames will be restricted to small transformations.

[5]During the learning of the parameters of $h$, we detach the gradients of the latent code $\mathbf{z}$ with respect to the parameters of the encoder $f$.

[6]Additionally, we forward the latent code through a neural projection head `proj` before computing similarity.

---

**Algorithm 1:** Implementing our Lie operator in a self-supervised learning model

    **Input: x, x'**

1 Encode $\mathbf{z} = f(\mathbf{x})$, $\mathbf{z}^g = f(\mathbf{x}')$ with the encoder $f$;

2 Infer $\hat{\mathbf{t}}$ as $\hat{\mathbf{t}} = h(\mathbf{z}, \mathbf{z}^g, \delta_{\mathbf{x},\mathbf{x}'})$;

3 Use the $\exp$ map to get $\hat{g} = \exp(\hat{\mathbf{t}}^\intercal \mathbf{L})$, where $\hat{g} \in \mathbb{R}^{m \times m}$;

4 Compute $\mathbf{z}^{\hat{g}} = \hat{g}(\mathbf{z})$. This is simple matrix-vector multiplication;

5 Jointly learn $f$, $h$, and $\mathbf{L}$ by minimizing:

$$\mathcal{L} = \lambda_{\text{ssl}}(L_{\text{ssl}}(\mathbf{z}) + L_{\text{ssl}}(\mathbf{z}^g)) + \lambda_{\text{lie}}L_{\text{Lie}}(\mathbf{z}^{\hat{g}}, \mathbf{z}^g) + \lambda_{\text{euc}}||\mathbf{z}^g - \mathbf{z}^{\hat{g}}||^2 + s(\delta_{\mathbf{x},\mathbf{x}'})||\hat{\mathbf{t}}||^2 \quad (8)$$

---

bring robustness to the transformation $g$. However, we control for this in our experiments using the SimCLR model with our Lie operator (Appendix E, see SimCLR Frames baseline). We experiment with a baseline that is simply encouraged to match pairs of rotated frames in the representation, and SimCLR Lie outperforms this baseline. Thus, this shows that viewing transformations as data augmentation is not sufficient for robust generalization.

To further encourage $\mathbf{z}^g$ and $\mathbf{z}^{\hat{g}}$ to match, we add an Euclidean-type constraint:

$$\lambda_{\text{euc}}||\mathbf{z}^g - \mathbf{z}^{\hat{g}}||^2 \quad (6)$$

The addition of the Euclidean loss helps to stabilize training, but using only the Euclidean loss (in place of our Lie loss) would not be enough to prevent the model from collapsing to mapping all transformed samples to the same point in latent space. Indeed, in the case of the MAE model, we perform ablations of our Lie model in Section 4.3 and show that all terms in the training objective are important for best performance with the MAE Lie model.

### 2.3   Learning a Lie operator in self-supervised representations

We use our Lie algebra model on top of two existing self-supervised (SSL) models, MAE and VICReg. In MAE, $f$ is trained to reconstruct a masked input image, while VICReg is trained to match pairs of samples obtained from the same image via some form of data augmentations (e.g. two crops) and uses Variance-Invariance-Covariance regularization to prevent collapse. In Appendix E we also experiment applying our Lie operator on the SimCLR (Chen et al., 2020) model, showing gains in the linear setting in particular. Our loss will balance between two components: the original SSL model loss $L_{\text{ssl}}$, applied to both samples of the pair separately, and the transformation loss $L_{\text{Lie}}$. Combining these constraints gives us our final loss, $\mathcal{L}$:

$$\mathcal{L} = \lambda_{\text{ssl}}(L_{\text{ssl}}(\mathbf{z}) + L_{\text{ssl}}(\mathbf{z}^g)) + \lambda_{\text{lie}}L_{\text{Lie}}(\mathbf{z}^{\hat{g}}, \mathbf{z}^g) + \lambda_{\text{euc}}||\mathbf{z}^g - \mathbf{z}^{\hat{g}}||^2 + s(\delta_{\mathbf{x},\mathbf{x}'})||\hat{\mathbf{t}}||^2 \quad (7)$$

The parameters $\lambda_{\text{ssl}}$ and $\lambda_{\text{Lie}}$ weight the two losses. Note that $L_{\text{ssl}}$ can be different given the SSL model backbone we use. In our experiment we deploy our Lie operator on top of the Variance-Invariance-Covariance Regularization model (VICReg, (Bardes et al., 2021)), and of the very recent masked autoencoders model (MAE, (He et al., 2021)).

This loss is minimized with respect to the parameters of the encoder $f$, the parameters of the network $h$ (multi-layer perceptron with two linear layers connected by a leaky ReLU) that infers the Lie algebra coordinates, and the Lie algebra basis matrices $\mathbf{L_k}$, $k \in \{1, \dots, d\}$ which are used to compute $\hat{g}$. The total loss is over all the instances ($N$ of them) and all possible pairs of transformed samples we have for each instance (if we have $P$ pose changes, this will be $P(P-1)$). During training, we estimate this loss for a minibatch by sampling instances first, and sampling a pair of samples for each instance without replacement. The full procedure is illustrated in Fig. 1 and in Algorithm 1.

## 3   Evaluating generalization across instances

Our goal is to evaluate how well models generalize variation across instances. We focus on robustness to pose changes, a common, natural variation of objects. We evaluate the learned representations'

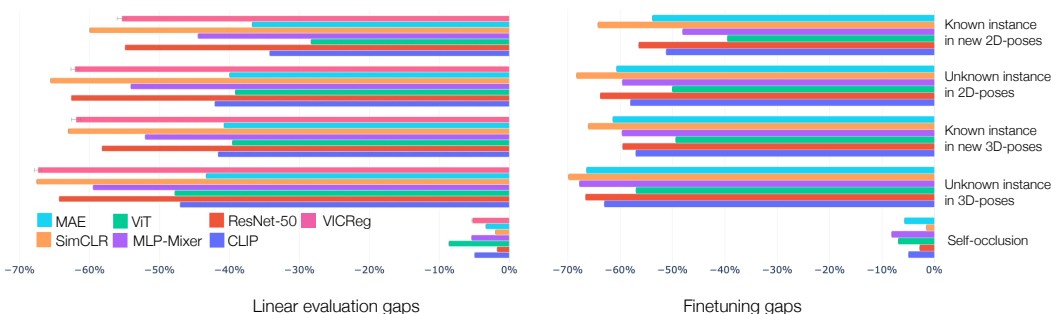

Figure 2: **SoTA image classification models fail to generalize across poses and instances.** Bar plots show the gap in top-1 accuracy relative to training instances with poses seen during training. Self-occlusion is the gap in accuracy between unknown instances in 3D and 2D poses.

robustness via standard linear evaluation or supervised finetuning protocols. Specifically, we measure classification top-1 accuracy across both *known* (seen during training) and *unknown instances* (not seen during training) in typical and new poses. As only a portion of instances (less or equal to 50% in our experiments) are seen varying, motivated by works showing the benefit of data reweighting for generalization challenges (Idrissi et al., 2022), we sample data at training so that the model sees as many images varying as non varying by upsampling the varying instances where needed.

In order to perform well on known instances in new poses, models must learn to recognize an object regardless of its orientation and constitutes our weakest generalization test. In order to perform well on unknown instances, however, models must be able to generalize both across instances and poses, a substantially more difficult task.

### 3.1 DATA VARIATION

To conduct our evaluation, we develop a controlled dataset of common objects such as buses, cars, planes, etc. based on realistic object models from Trimble Inc [7]. We render nearly 500,000 images of varying 2D and 3D poses uniformly distributed with a step size of 4 degrees. To simulate natural changes in pose we organize pose changes temporally into frames (i.e. into images). Each frame captures a 4 degree change in pose either in-plane rotation or across all three axes. In addition, we conduct experiments evaluating the Lie operator on the iLab 2M dataset following the preprocessing steps from Madan et al. (2021).

### 3.2 GENERALIZING POSE CHANGES IS CHALLENGING FOR SOTA VISION MODELS

We first assess state-of-the-art vision models' robustness to pose using a collection of supervised and self-supervised models (CLIP, ResNet-50, ViT-B/16, MLP Mixer, SimCLR, VICReg and MAE). All of these models rely on some form of data augmentation to provide invariance to nuisance features such as pose. We use ImageNet (Deng et al., 2009) pretrained embeddings for our models (with the exception of CLIP, which is pretrained on a large corpus of image-text pairs) (Radford et al.). We only show supervised linear evaluation for VICReg as it is mainly explored in the linear evaluation setting for the fully supervised task in Bardes et al. (2021).

While all models perform well on classification for known instances in typical poses, the same models have a considerable generalization gap to varying poses. In Figure 2, we show results for both linear evaluation and supervised finetuning classification gaps relative to typical poses of known instances. Strikingly, all models we evaluated exhibited large generalization gaps of 30-70% suggesting that even modern self-supervised models struggle to generalize both to new poses and to new instances. As such poor generalization can be problematic at best and dangerous at worst for deployed models, developing ways to improve generalization performance beyond simply augmenting the data is critical. In the next section, we introduce a new approach in which we encourage models not to be invariant to transformation, but rather to explicitly model and account for transformations.

---

[7]freely available under a General Model license

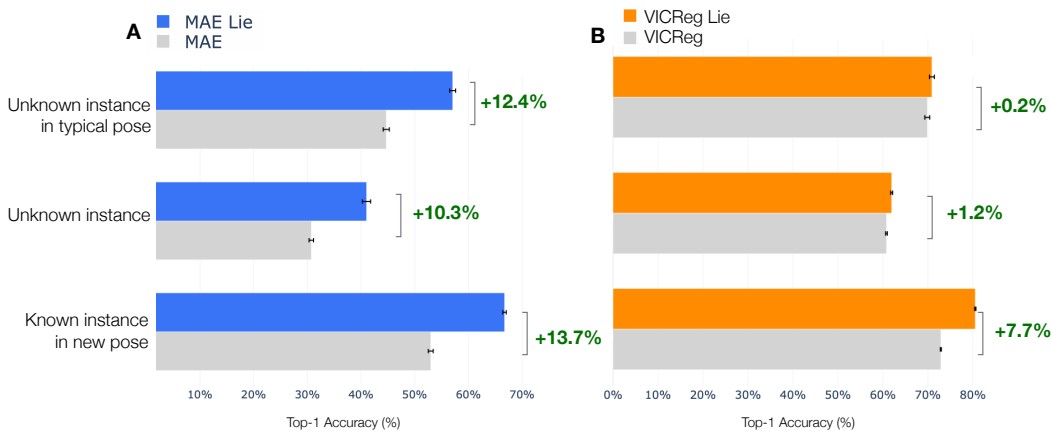

Figure 3: **Incorporating our Lie operator to MAE and VICReg improves generalization across poses and instances.** Bars indicate the mean top-1 accuracy and brackets indicating the standard error across seeds. In A., we show supervised finetuning for MAE Lie versus a standard MAE. In B., we show supervised linear evaluation for VICReg Lie versus VICReg as the original VICReg model is mainly explored in the linear evaluation setting for the fully supervised task (Bardes et al., 2021).

### 3.3 INCORPORATING A LEARNED LIE OPERATOR IN SELF-SUPERVISED LEARNING MODELS

To improve robustness, we incorporate a self-supervised learning phase where we structure representations using our Lie operator. During this phase, half of the instances are seen undergoing pose changes in 2D and 3D. We use the angle difference between frames as a proxy for frame distance (denoted $\delta$ in Section 2.1). In the case of rotation around all axes, we use the mean of three angles. We then evaluate the representations in the same manner as described above.

We implement our Lie operator in two SSL models: Masked Autoencoder (MAE) and Variance-Invariance-Covariance Regularization (VICReg). VICReg is a SSL method, which encourages similar representations among two views (generated via data augmentation) of an image. MAE is an SSL model whose learning objective is based on the reconstruction of a randomly masked input.

## 4 RESULTS

### 4.1 SSL MODELS WITH LIE OPERATORS ARE MORE ROBUST ACROSS INSTANCES AND POSES

We evaluate the robustness benefits of our Lie operator across several generalization settings. In Figure 3, we plot the performance of a MAE (**A**) and VICReg (**B**) models with and without our Lie operator. Encouragingly, we find explicitly modeling transformations with Lie groups (referred to as MAE Lie and VICReg Lie) improves robustness for all generalization settings. For known instances (those seen during training), incorporating our Lie operator improves Top-1 classification accuracy for new poses by $13.7\%$ for MAE Lie and $7.7\%$ for VICReg Lie. In the more challenging settings of unknown instances which are never seen during training, we still observe performance gains for VICReg Lie of $0.2\%$ for new instances in typical and $1.2\%$ for those in new poses. For MAE Lie, we observe substantial performance gains of $12.4\%$ for unknown instances in typical poses and $10.3\%$ for those in diverse poses. We note both MAE and VICReg Lie bring even more substantial gains for settings when data diversity is limited (see Sec. 4.2 Tables 1 and 2).

### 4.2 LIE OPERATOR ROBUSTNESS GAINS ARE SUSTAINED EVEN WITH LIMITED DIVERSE DATA

For our experiments so far, models have been exposed to a 50/50 mix of typical and atypical poses during supervised linear classification and finetuning. However, we want models to generalize across instances and to atypical poses even when presented with comparatively small amounts of diverse data. We therefore evaluated our models by performing supervised linear classification and finetuning with either 50%, 25%, or 5% of data featuring atypical poses (Tables 1 and 2). Notably, we observed that while overall performance decreased with fewer diverse instances, as we might expect, the

relative gains of the Lie models were largely consistent across settings, and if anything, were larger for settings with less diverse data. These results demonstrate the robustness of our approach to differing quantities of diverse data.

Table 1: **Assessing linear evaluation robustness to pose changes as the proportion of diverse instances seen during training varies.** Table reports linear evaluation top-1 accuracy as **mean** $\pm$ standard error (+ absolute difference, x relative multiple) to the baseline model.

| diverse proportion | known instance in new pose | | | | unknown instance | | |
| top-1 accuracy (%) | 5% | 25% | 50% | | 5% | 25% | 50% |
|---|---|---|---|---|---|---|---|
| **VICReg Lie** | 73.8 ± 0.2 **(+14.4, 1.24x)** | 75.0 ± 0.0 **(+5.8, 1.08x)** | 80.5 ± 0.1 **(+7.7, 1.11x)** | | 54.1 ± 0.2 **(+3.7, 1.07x)** | 59.7 ± 0.1 **(+3.1, 1.05x)** | 62.0 ± 0.2 **(+1.2, 1.02x)** |
| VICReg | 59.4 ± 0.0 | 69.1 ± 0.1 | 72.9 ± 0.1 | | 50.4 ± 0.1 | 56.7 ± 0.2 | 60.8 ± 0.2 |
| **MAE Lie** | 13.0 ± 0.2 **(+1.3, 1.11x)** | 13.3 ± 0.4 **(+2.3, 1.21x)** | 16.5 ± 0.3 **(+4.1, 1.33x)** | | 11.9 ± 0.5 **(+1.2, 1.12x)** | 11.5 ± 0.2 **(+1.2, 1.11x)** | 12.9 ± 0.4 **(+1.3, 1.11x)** |
| MAE | 11.6 ± 0.2 | 11.0 ± 0.4 | 12.4 ± 0.1 | | 10.6 ± 0.2 | 10.3 ± 0.4 | 11.6 ± 0.5 |

Table 2: **Assessing finetuning robustness to pose changes as the proportion of diverse instances seen during training varies.** Table reports finetuning Top-1 Accuracy as **mean** $\pm$ standard error (+ absolute difference, x relative multiple) to the baseline model. We do not experiment with VICReg finetuning as the original paper (Bardes et al., 2021) focuses on linear evaluation.

| diverse proportion | known instance in new pose | | | | unknown instance | | |
| top-1 accuracy (%) | 5% | 25% | 50% | | 5% | 25% | 50% |
|---|---|---|---|---|---|---|---|
| **MAE Lie** | 49.2 ± 0.5 **(+10.9, 1.28x)** | 60.4 ± **(+12.6, 1.26x)** | 66.7 ± 0.3 **(+13.7, 1.26x)** | | 28.4 ± 0.1 **(+8.2, 1.40x)** | 36.6 ± 0.1 **(+9.4, 1.35x)** | 41.0 ± 0.8 **(+10.3, 1.34x)** |
| MAE | 38.3 ± 0.5 | 47.8 ± 0.4 | 53.0 ± 0.5 | | 20.2 ± 0.1 | 27.2 ± 0.3 | 30.7 ± 0.4 |

### 4.3 ABLATIONS

In Table 2, compared to our MAE Lie, a model with only the SSL term (and no Euclidean or Lie loss), that is, the baseline MAE, suffers drops of $10.3\%, 9.4\%$, and $8.2\%$ in top-1 accuracy on unknown instances, respectively for the $5\%, 25\%$ and $50\%$ diversity settings.

In addition, we ran systematic ablations for each loss component in our MAE Lie model. We find a consistent decrease in performance across all three diversity settings. Compared to our MAE Lie model, a model without the Euclidean loss term ($\lambda_{euc} = 0$) suffers drops of $10.5\%, 9.08\%$, and $3.9\%$ in top-1 accuracy for unknown instances (when $50\%, 25\%$, or $5\%$ of instances vary in pose during training, respectively). Similarly, compared to our MAE Lie, a model with only the Lie loss (i.e. $\lambda_{euc} = \lambda_{ssl} = 0$) suffers a drop of $5.9\%, 5.01\%$, and $1.99\%$ in top-1 accuracy for unknown instances across the three diversity settings. This result illustrates how each loss term is critical to the overall performance of our proposed MAE Lie model.

## 5 MAE LIE TRANSFER ON REAL OBJECTS DATASETS

### 5.1 IMAGENET, ROT IMAGENET

Table 3: **Results comparing MAE Lie versus MAE Top-1 Accuracy for ImageNet, Rot ImageNet (random rotations)** after finetuning of 20k steps on the ImageNet training set.

| Top-1 Accuracy (%) | ImageNet Validation | Rot ImageNet Validation |
|---|---|---|
| MAE | 63.61 | 34.50 |
| MAE Lie | **67.55 (+3.94%)** | **37.12 (+2.62%)** |

In order to further demonstrate the benefit of our proposed Lie operator to standard benchmarks, we perform evaluation on ImageNet (Deng et al., 2009) and some of its variants. Rot ImageNet consists of the same validation images on which we apply random rotations to assess how well models handle pose changes. To do so, we finetune both the baseline MAE model and MAE Lie for 20k steps on ImageNet with a linear classification head. Table 3 reports the performance on the ImageNet and Rot ImageNet validation sets. We find MAE Lie outperforms the baseline for rotated ImageNet validation images by a considerable $+2.62\%$ in top-1 accuracy and the unaltered ImageNet validation set (by

+3.94% in top-1 accuracy) which contain pose along with other changing factors. Details on the training can be found in Appendix B.1.3.

## 5.2 ILAB DATASET

To demonstrate the applicability of our approach to other datasets, we also evaluate our MAE Lie model on the iLab 2M dataset. The iLab 2M dataset (Borji et al., 2016) is a natural image dataset consisting of 2M images of toy objects, from multiple categories, undergoing variations in object instances, viewpoints, lighting conditions, and backgrounds. We follow the preprocessing steps in Madan et al. (2021). We partition the training samples into a validation set (10%) and show results on the iLab test set. We conduct supervised finetuning, the standard evaluation protocol for MAE. We observe MAE Lie yields nearly a 3% improvement on the test set ($93.2\% \pm 0.45\%$ MAE Lie vs $90.6\% \pm 0.11\%$ MAE) without re-training our Lie operator on the iLab dataset.

## 6 RELATED WORK

**Modeling variation** Existing domain adaptation works (Zhou et al., 2020; Robey et al., 2021; Nguyen et al., 2021) learn transformations that map one domain to the other. In contrast to Ick & Lostanlen (2020); Hashimoto et al. (2017), we learn transformations in the latent space rather than image-space. Connor & Rozell (2020); Connor et al. (2021a) learn transformations in the latent space but uses a decoder that performs image reconstruction, which is known to be hard to apply to complex, large scale image data. In the context of classification, Benton et al. (2020) use the Lie algebra formalism to learn augmentations, but they require a priori knowledge of the augmentations. Tai et al. (2019) provides invariance to continuous transformations, but they only experiment on small scale data (MNIST (LeCun et al., 2010) and SVHN (Netzer et al., 2011)). We also depart from their work as we explicitly focus on the challenge of robustness to distributional shifts. Equivariant models (see Cohen & Welling (2016); Cohen et al. (2020) among others) and disentanglement (Higgins et al., 2018; Giannone et al., 2019) also aim to structure latent representations, but are mainly applied to synthetic, small scale data. Disentanglement also often requires costly pixel reconstruction. Spatial Transformer Networks (Jaderberg et al.) learn an instance-dependent affine transform for classification, resulting in behavior such as zooming in on the foreground object. Our goal differs as we focus on the generalization capabilities of various models to pose changes. Capsule networks (Sabour et al., 2017; Mazzia et al., 2021) were designed to better generalize to novel views, however, it is difficult for us to compare our approach to these as we require a strong pre-trained image model. To our knowledge, no such model using capsule networks exists, making a fair comparison very difficult.

**OOD Generalization failures** Alcorn et al. (2019); Engstrom et al. (2019) shed light on the failures of SoTA models on "unusual" examples that slightly vary from the training data, and Engstrom et al. (2019); Azulay & Weiss (2019) show that explicitly augmenting the data with such variations does not fix the problem. Indeed, Bouchacourt et al. (2021b) found that data augmentation does not bring the expected invariance to transformations, and that invariance rather comes from the data itself. Self-supervised models trained with InfoNCE theoretically recover the true data factors of variations (Zimmermann et al., 2021; von Kügelgen et al., 2021), but our experiments show that they do not generalize to distributional shifts.

## 7 DISCUSSION

We aim to improve the OOD generalization of SSL models by encouraging them to explicitly model transformations using Lie groups. Using a novel dataset to evaluate OOD generalization, we found that applying our Lie operator to MAE and VICReg leads to markedly improved performance for both models, enabling better generalization both across poses and instances, both on our synthetic data and on ImageNet and its variations for MAE. Our results highlight that data augmentation alone is not enough to enable effective generalization and that structuring operators to explicitly account for transformations is a promising approach to improving robustness. We focus on modeling pose changes, a common natural factor of variation in objects. However, Lie groups are capable of describing a variety of naturally occurring continuous transformations. Future work could extend the application of Lie operators to these settings, possibly even to model multiple factors of variation at once.

## 8 REPRODUCIBILITY STATEMENT

We made efforts to make all our work reproducible. In the main text, we describe every aspect of our learning objective and all the assumptions taken, with some background in Appendix A. Regarding our experiments, all training details are in Appendix B or in main text, and datasets descriptions are in main text or Appendix C. We will release the code to reproduce all experiments and generate the dataset based on realistic object models from Trimble Inc (freely available under a General Model license) upon acceptance.

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
