# OpenReview forum: "Robust Self-Supervised Learning with Lie Groups"
_ICLR.cc/2023/Conference — Submitted to ICLR 2023_

### Official Review · Reviewer_iZ5V · 2022-10-21

**Confidence:** 4
**Correctness:** 3
**Technical Novelty And Significance:** 3
**Empirical Novelty And Significance:** 3
**Recommendation:** 5

**Clarity, Quality, Novelty And Reproducibility:**

The paper is clearly written and introduces all relevant concepts in the main paper.

The concept of Lie groups has been used before in machine learning, but to my knowledge not for OOD robustness.

According to the "General Model License" of Trimble Inc., it is not allowed to "[...] use or incorporate such Model in any application, product, service, database or repository". This raises a reproducibility issue of the paper, since the used 500K images for training cannot be used by other researchers to reproduce the presented results.


**Strength And Weaknesses:**

Strengths:
+ The used mathematical concept of Lie groups/algebras is nicely explained
+ The four different loss components are explained in detail and ablation studies are performed to emphasize their usefulness
+ Newest models (like CLIP) are used to demonstrate the usefulness of this new approach

Weakness:
- It is not clear why 3D rotations should translate to group operations in the latent space. This two operations seem to be logically disconnected
- The paper is not clear about the (manifold) dimension of the Lie group. Depending on the latent space, this dimension might be quite high
- The proposed method uses different hyper-parameters for the four different loss components. Since these are not disclosed, it is difficult to reproduce the results.
- In Section 2, it is not clear why the scalar `t` is necessary. After all, the Lie algebra is a vector space and is closed under scalar multiplication

**Summary Of The Paper:**

The paper addresses the problem of OOD robustness with respect to pose changes. To this end, a Lie group transformation of latent representations is learned. This groups is parametrized via its Lie algebra in order to properly integrate it into a (vector space-driven) deep learning framework.

To train the network, pairs of images are used that show the same object at different poses.
Using these pairs, an SSL is performed by using a combination of four loss functions:
1. Standard SSL training loss
2. InfoNCE to measure the similarity of the latent representation (w.r.t. the algebra estimation)
3. Euclidean similarity between the latent representation
4. Task-dependent algebra-regularization

The proposed method is trained on a dataset of about 500K generated images based on models owned by Trimble Inc.

**Summary Of The Review:**

Overall, I like the idea presented in this paper. There are minor concerns like the Lie group's dimension.

My main concern is with respect to the reproducibility with stems from the license of the used data.
Therefore, I see the paper marginally below the acceptance threshold.

If the reproducibility issue can be resolved, I am happy to raise my vote.

---

> ### Author Response · Authors · 2022-11-11
> **Response**
>
> We appreciate the reviewer's thoughtful suggestions and attention to reproducibility, which we've made several changes to improve thanks to these suggestions. We also appreciate the reviewer noted the novelty of our approach in applying Lie algebras to learn continuous transformations to improve OoD robustness. We address specific comments below and would be glad to further clarify any lingering questions.
>
>  > "It is not clear why 3D rotations should translate to group operations in the latent space."
>
> We impose a structure on the latent space to facilitate generalization by ensuring variation in the input space corresponds to predictable operators matching the structure of rotations. This formulation closely follows the standard definition of equivariance with matching group structures between input and latent space [3]. In this case, SO(3) or special orthogonal group has a well-defined structure described by 3 Lie algebra matrices or SO(2) for 2D rotations described by 1 Lie algebra matrix, which we learn via backpropagation.
>
> > "The paper is not clear about the (manifold) dimension of the Lie group. Depending on the latent space, this dimension might be quite high"
>
> We note SO(3) this along with many other common transformations are low-dimensional. Transformations such as translations, shear, and scaling are known to be low dimension (all affine 2D transformations have dimension 6) (see for example [4]) making Lie algebra operators an appealing, computationally efficient approach to modeling transformations.
>
> > "The proposed method uses different hyper-parameters for the four different loss components. Since these are not disclosed, it is difficult to reproduce the results."
>
> We agree disclosing hyper-parameter choice is critical to reproducibility. As noted in the original submission’s Appendix B.1.2,  we specify hyperparameter choices as well as the ranges used to perform hyperparameter sweeps. While we can not include the data due to both license and their sheer size, we further provide code to reproduce all experiments in PyTorch in a zip-file. We will also open-source this code following a decision on the paper.
>
> > why the scalar t is necessary. After all, the Lie algebra is a vector space and is closed under scalar multiplication”
>
> As you keenly noted, the Lie algebra forms a vector space, which is key to our method’s ability to learn the transformations via backpropagation. The t parameter specifies the constant coefficients used to specify an element in the Lie algebra (i.e., coordinate of the transformation in the Lie algebra) from the Lie algebra basis matrices. Intuitively for pose, this constant t would tell us the degree of rotation for example. Furthermore, as we note in future directions the scalar can also express the extent to which multiple transformations simultaneously vary—when working with say video data.

---

> > ### Author Response · Authors · 2022-11-11
> > **License and reproducibility**
> >
> > > "My main concern is with respect to the reproducibility with stems from the license of the used data."
> >
> > We admire the reviewer’s attention to reproducibility and absolutely agree reproducibility is critical to facilitating research. We’ve provided code to reproduce all experiments. We also provide code to generate data in the precise manner used for experiments. This code can also be extended or used with any other set of 3D objects. Finally, we’ve also included pseudo-code in Appendix C to illustrate the method more clearly for easy adoption in future research. We hope this along with our response fully address any lingering questions.
> >
> > Specifically regarding the concerns around the license, we identified the license for Timble Inc. allows use for non-commercial research. Specifically, 3D Warehouse is available under the General Model license (https://3dwarehouse.sketchup.com/tos/#license), which allows (see Authorized uses):
> >
> > `Creating derivative works of Models (“Creations”), including by substantially modifying geometry, color, or other attributes of the Models, provided that if you upload any Creation to the 3D Warehouse, you will be a Developer under the 3D Warehouse Terms of Use and such Creation shall constitute a “Model” under the Terms of Use;
> > Incorporating or including Models and Creations into a larger work or a deliverable for a third party (“Combined Work”), provided that the Combined Work includes substantial additional content to the original Model;
> > Distributing Models, Creations, and Combined Works to third parties for your business purposes (including for commercial purposes);
> > Making a reasonable number of copies of the Models, Creations, and Combined Works in connection with the above uses; and
> > Using the Models as Trimble may approve from time to time in its sole discretion.`
> >
> > We appreciate the Reviewer’s attention to  “Google geolocated models”, which are defined in the license as:
> >
> > `“Google Geolocated Models” are Models Distributed through 3D Warehouse by Google Inc. (or by Trimble on Google’s behalf) that constitute a 3D model of an object, building or other structure with a fixed location that is tagged or otherwise associated with the actual geolocation of such object, building or other structure.`
> >
> > We confirm we do not use any objects under this definition in our set of objects. Similarly note that Trimble Inc. objects have been used in recent CVPR 2022 papers such as [1] and [2]. We kindly ask the Reviewer to let us know if this clarifies license usage regarding reproducibility. Finally, note that the code we share can also be extended or used with any other set of 3D objects.
> >
> > We hope this information regarding the license as well as code, hyperparameter choices, etc. address any remaining concerns regarding reproducibility.
> >
> > [1] Pre-train, Self-train, Distill: A simple recipe for Supersizing 3D Reconstruction , Kalyan Vasudev Alwala, Abhinav Gupta, Shubham Tulsiani CVPR 2022 https://shubhtuls.github.io/ss3d/
> >
> > [2] Learning 3D Object Shape and Layout without 3D Supervision, Georgia Gkioxari, Nikhila Ravi, Justin Johnson CVPR 2022 https://gkioxari.github.io/usl/
> >
> > [3] Cohen, Taco S., and Max Welling. “Group Equivariant Convolutional Networks.” arXiv, June 3, 2016. https://doi.org/10.48550/arXiv.1602.07576.
> >
> > [4] Benton, Gregory, Marc Finzi, Pavel Izmailov, and Andrew Gordon Wilson. “Learning Invariances in Neural Networks.” ArXiv:2010.11882 [Cs, Stat], December 1, 2020. http://arxiv.org/abs/2010.11882.

---

> > ### Comment · Reviewer_iZ5V · 2022-11-23
> > **Comment on Response**
> >
> > I would like to thank the authors for their response related to the Lie algebras. The following unclear points still remain:
> > * Separation between factor `t` and the algebra. Since the algebra is a vector space, the parameter `t` seems to be superfluous. The authors did not address this aspect in their algorithmic design
> > * The paper claims a Lie algebraic approach in the latent space (which might have a large dimension). At the same time, equivariance is only discussed (both in the paper and the authors' response) with respect to SO(3). These inconsistencies are not resolved by the authors' answers.

---

> > > ### Author Response · Authors · 2022-11-23
> > > **Further clarification on the role of t and dimension**
> > >
> > > We’d be glad to clarify these points.
> > >
> > > - The algebra allows us to model the type of the transformation with a few number of learnable matrices. This is akin to learning that the transformation is for example rotation. Then for a given pair of images, we need to infer the degree by which they differ for the given transformation, in our case, we do this by inferring the transformation’s coordinates in the Lie algebra, denoted t, see in Equation 2 in our paper. This is akin to inferring the angle of rotation for a given pair of images. We describe the role of t in Section 2.1. This allows us to learn a single set of basis matrices, for all the pairs corresponding transformations, then combine them (with exponential of their linear combination, each matrix weighted by their corresponding t, see Equation 1 in our paper) to specify how a given pair of images differ. While we can’t further update the paper at this stage, we’d be glad to update the text for clarity based on your suggestions upon publication.
> > > - We did our best to re-read our previous remarks, but are unable to identify the points leading to confusion. Could you please clarify where you see any inconsistencies? We’d be glad to address this.
> > > Please note at most the latent space is of dimensionality 2048 across the models we use. Importantly, the dimension of the Lie algebra of the group, which is defined by the number of basis matrices, is low dimensional. Most common transformations such as translations, shear, and scaling are known to be low dimension: all affine 2D transformations have dimension 6, i.e. can be represented by only 6 matrices. This is where the use of the coordinates t appears, to combine these 6 matrices to obtain every element of the group (see for example [4]) making Lie algebra operators an appealing, computationally efficient approach to modeling transformations. Finally, regarding equivariance while we use SO(3) as a prototypical example for clarity, the notion of equivariance within our method applies to any continuous transformation with the structure of a Lie group.

---

> ### Author Response · Authors · 2022-11-17
> **Have we addressed your questions regarding reproducibility?**
>
> Please let us know if the additional information regarding hyperparameters, codebase, and our response regarding the license sufficiently addresses your concerns regarding reproducibility. If so, we'd appreciate it if you'd consider raising your score. We remain at your disposal to answer any outstanding questions. Thank you.

---

> > ### Comment · Reviewer_iZ5V · 2022-11-23
> > **Comment on Reproducibility**
> >
> > The problems with respect to the used data set and the related problems of reproducibility are not addressed.
> >
> > As the authors pointed out, the license clearly states:
> > ```
> > Using the Models as Trimble may approve from time to time in its sole discretion.
> > ```
> >
> > It is therefore not clear if the experiments are still reproducible depending on how Trimble Inc. is changing their business model in the future.

---

> > > ### Author Response · Authors · 2022-11-23
> > > **Confirmation license allows for reproducibility**
> > >
> > > We thank Reviewer iZ5V for the reply. We believe there is a misunderstanding of this sentence given the context of paragraph 1.B in the license https://3dwarehouse.sketchup.com/tos/#license . Section 1.B enumerates a list of possible use cases, which firmly encompass reproducibility for research purposes. Use case v. which you draw our attention to, just implies Trimble inc. may approve further use cases beyond what’s covered in Section 1.B. Importantly, please note that as we mention in our reply, there are several existing works published at other AI conferences such as CVPR 2022 using the same set of models. Finally, please note that we consulted with the Legal department of our institution and received approval to pursue using these models in our open-research work before submitting to the conference. Therefore, we do not see any issues with reproducibility given the license, ample previous work using the same models, and legal guidance.

---

### Official Review · Reviewer_wpyw · 2022-10-24

**Confidence:** 4
**Correctness:** 2
**Technical Novelty And Significance:** 2
**Empirical Novelty And Significance:** 2
**Recommendation:** 3

**Clarity, Quality, Novelty And Reproducibility:**

The writing of this paper is not clear. The proposed method only uses the idea of Lie group and contrastive learning. The novelty of this paper is limited. Besides, the proposed method contains multiple operations and is complex. It is better to give the code to reproduce the results.



**Strength And Weaknesses:**


Using group theories to improve OOD robustness is an interesting idea.


1. The motivation of this paper is not clear. In Introduction Section, the authors directly indicate that self-supervised models own limited OOD robustness. However, I am not clear why self-supervised models are with limited OOD robustness. The authors should give more interpretations.

2. Some indications in this paper are not accurate. For example, the authors indicate that for image data, augmentations can only be applied to the pixels themselves. However, to the best of my knowledge, there exist some feature-level augmentation methods. Besides, the authors should further interpret why augmentation fails to generalize to novel objects.

3. The writing of the method section is very chaotic. The proposed method contains multiple operations. However, the authors do not introduce the motivation of these operations clearly. Besides, it is better to give some theory analyses for the proposed method.

4. In Table 1 and 2, this paper only compares MAE and VICReg. The authors should evaluate the effectiveness on more methods and tasks. The proposed method involves multiple operations. The authors should make sufficient ablation experiments. Finally, the authors should give more training details, e.g., the training curves, and feature-level visualization analyses, which is beneficial for further understanding the proposed method.



**Summary Of The Paper:**

Though deep learning has achieved many advances, many existing methods suffer from poor generalization performance due to the domain-shift impact. To this end, this paper focuses on improving the robustness of self-supervised methods. Specifically, the authors choose matrix Lie groups to model continuous transformations and employ contrastive learning to enhance the generalization. In the experiments, the proposed method is evaluated on multiple datasets.




**Summary Of The Review:**

The writing of this paper is not clear. The proposed method only uses the idea of Lie group and contrastive learning. The novelty of this paper is limited. Besides, the proposed method contains multiple operations and is complex.

---

> ### Author Response · Authors · 2022-11-11
> **Clarifications and Improvements**
>
> We have made several improvements to the clarity of the method including adding a pseudo-code in Appendix C, Listing 1 as well as provide PyTorch code for the method, baselines, dataset, and experimental configurations. We also note other reviewers found the method to be quite novel as this work combines the formalism of Lie groups to learn without prior knowledge of the variation in the data. We address specific points below, particularly highlighting the simplicity of the method’s implementation:
>
> 1. > “ The motivation of this paper is not clear…I am not clear why self-supervised models are with limited OOD robustness.”
>
> Vision models are widely known to suffer in out-of-distribution (OoD) generalization. Standard benchmarks such as ImageNet-A or ObjectNet show a considerable drop in performance for out-of-domain samples. For pose specifically,  [1, 2], which we cite in our original submission, illustrate the brittleness with respect to pose changes. There is a substantial body of previous work illustrating that while self-supervised learning is slightly more robust, it still suffers considerably with respect to OoD such as [3,4, 5] among others. We also show in Section 3.2, SoTA self-supervised learning methods such as VICReg and MAE drop by more than 40% in top-1 accuracy when pose varies for held-out and even objects seen during training, further highlighting a major defect in SSL models’ ability to generalize OoD.
>
> 2. > “the authors indicate that for image data, augmentations can only be applied to the pixels themselves. However, to the best of my knowledge, there exist some feature-level augmentation methods.”
>
> We agree that while data augmentation is most often applied on pixel space, it can be at times applied in the manifold e.g. manifold Mixup [6]. Thank you for pointing this out, we have softened our wording in the paper accordingly in the introduction.
> “Besides, the authors should further interpret why augmentation fails to generalize to novel objects”
> While we agree that intuitively data augmentation is expected to provide generalization by bringing invariance to variations while keeping information about the object class, in our experiments in the original submission, we find otherwise. Our baseline self-supervised learning models are all trained using data augmentation (where we show the model pose changes during training). We show our method surpasses data augmentation in terms of generalization to known, unknown instances by more 10% for MAE for example. We interpret this as the fact that data augmentation is a proxy for enforcing invariance, applied only on training, seen objects, with no guarantee to provide the right type of robustness when theoretical assumptions break. This has been noted in [7] that studies the isolation of class information thanks to data augmentation, and finds that “while class should, in principle, always be invariant across views (i.e., content)—when using only crops, colour distortion, or rotation, g appears to encode shortcuts” where g is their encoder g: X -> Z. Furthermore, as we highlighted in our original submission, earlier works such as [1] showed augmentation can improve performance only for known objects but often fails to generalize to novel (or held-out) objects. We hope this clarifies the motivation behind the method and novelty with respect to existing work.
>
> 3. > "The writing of the method section is very chaotic. The proposed method contains multiple operations...it is better to give some theory analyses for the proposed method."
>
> The approach is motivated by the formalism of Lie groups, a theoretically grounded structure for modeling continuous transformations. In our case, the method is based on an important theoretical correspondence between the Lie algebra generator matrices (that form a learnable vector space) and the Lie group. The method is quite simple: we learn a set of basis matrices which we apply to exponential function to model continuous transformations in latent space. We also point out both Reviewer riZ5V and Reviewer 1hzj found the paper well-written, specifically noting “the used mathematical concept of Lie groups/algebras is nicely explained” and that “the four different loss components are explained in detail and ablation studies are performed to emphasize their usefulness.” We also color coded the equations to clearly delineate the note role of each loss term. Nevertheless, we agree clarity is important. Therefore, we now also include pseudo-code to illustrate the clarity of the method in Appendix Section C Listing 1 as well as theoretical intuition in the method section, Section 2. We hope this makes clear the sound mathematical motivation for the Lie operator and simplicity. Please do let us know if specific aspects are unclear.

---

> > ### Author Response · Authors · 2022-11-11
> > **Continued**
> >
> >
> > 4. > "The authors should make sufficient ablation experiments”
> >
> > We would like to emphasize that in the original submission we ran systemic ablations of each term of our learning objective in the MAE Lie model in Section 4.3 explicitly showing the relevance of every component for best generalization performance with MAE Lie.
> >
> > > “The authors should evaluate the effectiveness on more methods and tasks.”
> >
> > Note that in the original submission, we additionally report our model on top of the SimCLR architecture in Appendix E. By using our Lie operator on MAE, VICReg and SimCLR, we believe we cover different type of losses (contrastive for SimCLR, variance regularization for VICReg and auto-encoder type losses for MAE) and training settings of state-of-the-art self-supervised models. Finally, we report MAE Lie performance on realistic datasets (ImageNet, RotImageNet, and iLab). Together, this forms a comprehensive suite of models and tasks we confidently believe validate our approach.
> >
> > > "Finally, the authors should give more training details, e.g., the training curves, and feature-level visualization analyses, which is beneficial for further understanding the proposed method.”
> >
> > Please also note that we reported in the original submission all training details (hyperparameters, architectures, etc) in Appendix B. We thank you for the suggestion of adding training curves, we have added training curves for VICReg Lie in Appendix F (Figure A4 and A5) both for the self-supervised finetuning phase and linear evaluation, and we have uploaded our code as well as pseudo code in Appendix C. While an analysis with visualization of what the network learned is an interesting line to follow, we believe this is a direction of future work, with the use of consistent and relevant metrics and tools that is an ongoing topic of research by itself.
> >
> > [1] Alcorn, Michael A., Qi Li, Zhitao Gong, Chengfei Wang, Long Mai, Wei-Shinn Ku, and Anh Nguyen. “Strike (with) a Pose: Neural Networks Are Easily Fooled by Strange Poses of Familiar Objects.” ArXiv:1811.11553 [Cs], April 18, 2019. http://arxiv.org/abs/1811.11553.
> >
> > [2] Madan, Spandan, Timothy Henry, Jamell Dozier, Helen Ho, Nishchal Bhandari, Tomotake Sasaki, Frédo Durand, Hanspeter Pfister, and Xavier Boix. “When and How Convolutional Neural Networks Generalize to Out-of-Distribution Category–Viewpoint Combinations.” Nature Machine Intelligence 4, no. 2 (February 2022): 146–53. https://doi.org/10.1038/s42256-021-00437-5.
> >
> > [3] Shi, Yuge, Imant Daunhawer, Julia E. Vogt, Philip H. S. Torr, and Amartya Sanyal. “How Robust Are Pre-Trained Models to Distribution Shift?” arXiv, June 17, 2022. https://doi.org/10.48550/arXiv.2206.08871.
> >
> > [4] Geirhos, Robert, Kantharaju Narayanappa, Benjamin Mitzkus, Matthias Bethge, Felix A. Wichmann, and Wieland Brendel. “On the Surprising Similarities between Supervised and Self-Supervised Models.” arXiv, October 16, 2020. https://doi.org/10.48550/arXiv.2010.08377.
> >
> > [5] Hendrycks, Dan, Mantas Mazeika, Saurav Kadavath, and Dawn Song. “Using Self-Supervised Learning Can Improve Model Robustness and Uncertainty.” ArXiv:1906.12340 [Cs, Stat], October 29, 2019. http://arxiv.org/abs/1906.12340.
> >
> > [6] Manifold Mixup: Better Representations by Interpolating Hidden States, Vikas Verma, Alex Lamb, Christopher Beckham, Amir Najafi, Ioannis Mitliagkas, David Lopez-Paz, Yoshua Bengio, ICML 2019
> >
> > [7] Self-Supervised Learning with Data Augmentations Provably Isolates Content from Style, Julius Von Kügelgen, Yash Sharma, Luigi Gresele, Wieland Brendel, Bernhard Schölkopf, Michel Besserve, Francesco Locatello, NeurIPS 2021.

---

### Official Review · Reviewer_1hzj · 2022-10-25

**Confidence:** 4
**Correctness:** 4
**Technical Novelty And Significance:** 3
**Empirical Novelty And Significance:** 4
**Recommendation:** 8

**Clarity, Quality, Novelty And Reproducibility:**

This paper is well-written and presents a novel perspective to enable self-supervised models to generalize variation across objects.

**Strength And Weaknesses:**

Strength: This paper shows great potantial of learning the transformation in laten space induced by changes in pose to enable self-supervised models to generalize variation across objects. The perspective is novel and ablation study shows promising results.

Weaknesses: In Section 2.1, there is no description about how to decide the dimention d of the Lie algebra. In my understanding, there should be a way to specify the dimention, or it is the degree of freedom of the transformation matrix in Li group. Is my understanding correct?

**Summary Of The Paper:**

This paper proposes to apply the formalism of Lie groups to capture continuous transformationsto improve models robustness to distributional shifts. Specifically, it structures the representation of corresponding vector space of the assumed Lie group by learning some basis metrics and then constructs their Lie operator by using this vector space structure. The general idea is impressive, and the experimental results are promising.

**Summary Of The Review:**

In general, the presented method is novel and effective, which is also well-supported by the experimental results.

---

> ### Author Response · Authors · 2022-11-11
> **Response**
>
> We’re glad you appreciate the paper is well-written, our experiments are thoroughly conducted, and our idea is novel. We thank you for the excellent suggestion to clarify the role of the dimension of the Lie group in Section 2.1. You’re precisely correct: this dimension expresses the degrees of freedom or number of Lie algebra generators required to model the transformations. Specifically, it corresponds to the dimension of the Lie algebra as a vector space. In our preliminary experiments, we have cross-validated between a dimensionality of d=1 and d=5 and found it did not seem to alter the results, hence we use d=1 in our reported results. A higher dimension d of the Lie algebra allows for more expressivity to model more complex transformations. We expect that when multiple factors are varying (e.g. pose, size, location) a larger value of d is needed, which we would like to investigate in future work. We’ve updated the Section to specify with a precise description of this dimension d in Section 2.1.

---

### Author Response · Authors · 2022-11-11
**Thank you for the insight comments**

We thank the reviewers for their thorough reading and insightful comments. Thanks to these suggestions, we’ve improved the quality of our submission. We briefly note a majority of reviewers (1hzj and riZ5V) found the paper clear and well written. Specifically, riZ5V states “the used mathematical concept of Lie groups/algebras is nicely explained” and that “the four different loss components are explained in detail and ablation studies are performed to emphasize their usefulness”. Reviewer 1hzj notes the idea is impressive and results are promising, while Reviewer riZ5V also recognizes the novelty of the use of Lie groups to tackle out-of-distribution (OoD) robustness. We reply to each reviewer’s questions under their respective comment.

To address the primary concerns raised by Reviewers wpyw and iZ5V around implementation/reproducibility of the method, we’ve shared a zip file of our code repo with a full implementations (of our method, datasets, and experiments), included further clarifying text regarding the method in Section 2.1, and added a new PyTorch pseudo-code algorithm in Appendix C to ensure the research community can build on and extent this promising, theoretically grounded approach to improving robustness. To ease readability, we highlighted any changes both in the main paper and supplementary in blue.

We’d be glad to answer any further questions or engage in related discussions.

---

### Decision · Program_Chairs · 2023-01-20

**Decision:**

Reject

**Justification For Why Not Higher Score:**

Although the paper performs valuable explorations on the topic, AC thinks the reasons to reject slightly outweigh the reasons to accept at the current form.

**Justification For Why Not Lower Score:**

N/A

**Metareview: Summary, Strengths And Weaknesses:**

This paper proposes a method for improving the robustness to distribution shifts of self-supervised methods by modeling continuous transformations via Lie groups. A majority of reviewers are negative toward acceptance of this paper. There is only one positive reviewer, but it is hard to say the positive review is strong enough to support the acceptance and did not show strong support in the AC-reviewer discussion phase upon other reviewers' concerns. Although AC somewhat understands the authors' complaint about the reproducibility issue concerning dataset license but does not think the authors succeed in resolving other concerns of negative reviewers. In particular, AC agrees that "the authors should evaluate the effectiveness on more methods and tasks", "the authors should make sufficient ablation experiments", and "the proposed method only uses the idea of Lie group and contrastive learning," pointed out by reviewers. In addition, the authors claim that the proposed method can model continuous data transformations such as pose changes, but AC thinks that it should be further justified and investigated whether it actually does so. Moreover, AC thinks the difference between Lie and contrastive loss also should be further discussed to clarify where the benefits come from.